# Preparation and Herbicidal Evaluation of Butyl Hydroxybenzoate Emulsion

**DOI:** 10.3390/plants14193041

**Published:** 2025-10-01

**Authors:** Tianqi Wang, Haixia Zhu, Lijuan Bao, Suifang Zhang, Yongqiang Ma

**Affiliations:** 1Academy of Agriculture and Forestry Sciences, Qinghai University, Xining 810016, China; ys230951320689@qhu.edu.cn (T.W.); 18893076204@163.com (L.B.); 17389539023@163.com (S.Z.); 2Xining Observation and Experimental Station of Crop Pests, Ministry of Agriculture, Xining 810016, China; 3Key Laboratory of Comprehensive Management of Agricultural Pests in Qinghai Province, Xining 810016, China

**Keywords:** butyl hydroxybenzoate, preparation of emulsion, phytotoxicity test, safety evaluation, field effect evaluation

## Abstract

In order to develop a new environmentally friendly microbial herbicide for the field of weed control, this study used the metabolite butyl hydroxybenzoate (BP) of the HY-02 strain of *Alternaria* as the research object. The BP emulsion formula was determined to be a mixture of BP, methanol, and Tween-20 in a ratio of 1:1:2 g/mL. The seed germination inhibition effect, the phytotoxicity of living plants, crop safety, and the field effect of the emulsion were studied. Research has found that adding 0.75% BP emulsion to the seed culture medium inhibits the germination of weed seeds such as *Amaranthus retroflexus* L., *Malva verticillata* L. var., and *Chenopodium album* L. While *Brassica campestris* L. seeds were unaffected, *Triticum aestivum* L and *Hordeum vulgare* L. stem and leaf growth were inhibited. *Cucumis sativus* L., *Lactuca sativa* L. var. *asparagina*, *Spinacia oleracea* L., and *Capsicum annuum* L. seeds are significantly inhibited, with germination rates below 20%. We sprayed 0.75% BP emulsion onto live potted plants; among the weeds, the incidence of *Amaranthus retroflexus* L., *Lepyrodiclis holosteoides, Thlaspi arvense* L, *Galium spurium* L., *Malva verticillata* L. var. *Crispa*, *Chenopodium album* L., and *Avena fatua* L reached 100%. The *Pisum sativum L.* and *Triticum aestivum* L. crops were not affected (NS), and they had slight plant height inhibition and slight susceptibility (LS) to highland *Hordeum vulgare* L. and peppers. They were highly phytotoxicity to *Cucumis sativus* L. and *Spinacia oleracea* L. Some plant leaves became infected and died, with incidences of 85% and 82%, respectively. The field experiment showed that after diluting the BP emulsifiable concentrate, the seedling stage spray was inoculated into the *Triticum aestivum* L. field, and it was found that the BP emulsifiable concentrate at the concentration of 1.00%~0.75% had a herbicidal effect on weeds such as *Chenopodium album* L., *Elsholtzia densa Benth*, and *Amaranthus retroflexus* L. in the *Triticum aestivum* L. field, and it was safe for *Triticum aestivum* L. crops in the field. These results indicate that BP emulsion could be developed into a new environmentally friendly microbial herbicide for field application in grass (*Triticum aestivum* L. and *Hordeum vulgare* L.) crops. At the same time, BP’s excellent antibacterial, low-toxicity, hydrolysis, and other effects can promote diversification in herbicide development.

## 1. Introduction

The new sustainable development of agriculture has established strict requirements for herbicides to become environmentally friendly and pollution-free. Therefore, microbial herbicides with low toxicity, environmental compatibility, low degradability, weed resistance, minimal pesticide residue, and other advantages have become a higher pursuit for people [1]. Microbial herbicides are divided into live microbial herbicides and microbial-derived herbicides [2]. Compared with chemical herbicides and live microbial herbicides, microbial herbicides have unique chemical structures, good compatibility with the environment, strong specificity, and high selectivity; are easily stored; and are conducive to formulation processing, use, development, and synthesis costs [3]. Given the above advantages, the formulation of microbial herbicides has become increasingly diversified with the exploration of researchers, such as powders, emulsions, pastes, etc. An emulsion concentrate (EC) is easy to use, can save a lot of time and cost, and has high stability and flowability to protect active ingredients from factors such as moisture and light, thereby extending the shelf life. After adding surface activity, it can promote the diffusion and retention of EC on plant surfaces, achieving better coverage and more consistent control [4].

In recent years, breakthrough progress has been made in microbial herbicides, and fungi account for a large proportion of the development and research of microbial herbicides [5]. Many research results have confirmed that many toxins produced by *Alternaria* have the potential to be developed as microbial herbicides [6]. The TeA toxin extracted from *Alternaria* can inhibit the photosynthesis of *Eupatorium adenophorum*, causing toxic effects on it. AAL toxins in *Alternaria* have strong control effects on *Solanum nigrum*, *E. adenophorum*, and *Datura stramonium* [7]. The solanomycin obtained from Streptomyces actinomycetes has a controlling effect on *Echinochloa crusgalli* L. and *Digitaria san guinalis* L., but it has no toxic effect on cultivated crops such as *Oryza sativa* L [8]. In the research process of microbial herbicides, most researchers have conducted extensive research on the isolation, identification, and mechanism of action of toxins, but there is relatively little research on the development of toxins.

Butyl paraben (BP), also known as nipagin butyl ester or nipagin, has the chemical name 4-hydroxybenzoate butyl ester [9]. BP is an ester compound that contains a portion of para hydroxybenzoic acid and butyl ester in its structure, and it has good stability and broad-spectrum antibacterial properties [10,11,12]. In daily life, due to its characteristics towards hydroxybenzene, BP can effectively prevent the growth of microorganisms and ensure the stability of product quality during storage [13]. Therefore, BP is often used as an effective additive in various formulations such as sunscreen, skincare products, and disinfectants [14]. In preliminary research, the active substances in the fermentation filtrate of *Alternaria alternata* HY-02 were separated and purified by column chromatography, thin layer chromatography, and high-performance liquid chromatography to obtain the monomeric compound BP. After a literature review, this compound has not been reported in the field of weed control and biocontrol. Therefore, this study mainly evaluates the weed control potential of BP, providing more alternative resources for the development and research of microbial herbicides. At the same time, BP’s excellent antibacterial and other effects can also promote the diversification of the prospects in the field of herbicides.

## 2. Materials and Methods

### 2.1. Test Drugs and Reagents

The following drugs and reagents were used: dimethyl sulfoxide, anhydrous ethanol, methanol, xylene, dichloromethane, and n-butanol (analytical purity; Tianjin Fuyu Fine Chemical Co., Ltd., Tianjin, China); Tween 20 and Tween 80 (Tianjin Zhiyuan Chemical Reagent Co., Ltd., Tianjin, China); agricultural milk 1601 OV-5, fatty alcohol polyoxyethylene ether phosphate ester, and OF-4816-B (Xingtai Xinlanxing Technology Co., Ltd., Xingtai, China); glucose (analytical purity; Hongyan Reagent Factory in Hedong District, Tianjin, China); agar powder (analytical grade; Beijing Aoboxing Biotechnology Co., Ltd., Beijing, China); and a 9 cm culture dish (China National Pharmaceutical Group Chemical Reagent Co., Ltd., Shanghai, China).

### 2.2. Test Weeds

The following weeds were used: *Chenopodium album* L., *Elsholtzia densa Benth*, *Malva verticillata* L. var. *Crispa*, *Thlaspi arvense* L., *Amaranthus retroflexus* L., *Galium spurium* L., *Lepyrodiclis holosteoides*, and *Avena fatua* L.

### 2.3. Test Crops

The following crops were used: *Capsicum annuum* L., *Cucumis sativus* L., *Spinacia oleracea* L., *Lactuca sativa* L. var. *asparagina, Pisum sativum* L., *Hordeum vulgare* L., *Triticum aestivum* L., and *Brassica campestris* L.

### 2.4. Selection of Solvents for Butyl Hydroxybenzoate Emulsion

Six 5 mL cryovials were taken, and 1 g of raw BP was weighed. Then we added 1 mL of each of the 6 organic solvents (dimethyl sulfoxide, anhydrous ethanol, methanol, xylene, dichloromethane, and n-butanol) into the 6 cryovials. The mixture was shaken at room temperature to promote dissolution. If dissolution was incomplete, add solvent dropwise and shake to dissolve. When the solvent was added to 5 mL, it still could not dissolve completely, and the reagent was discarded [15]. Finally, considering the solubility and cost factors of six organic solvents, the most suitable organic solvent was selected as the emulsion solvent.

### 2.5. Formulation and Determination of BP Emulsifiable Concentrate

#### 2.5.1. Screening of Emulsifiers

At room temperature, we selected Tween 20, Tween 80, agricultural milk 1601 OV-5, and six different emulsifiers and solvents with the best solubility, namely fatty alcohol polyoxyethylene ether phosphate ester and OF-4816-B. They were prepared according to the formula of 1.0 g/mL for the original drug, 1.5 g/mL for the emulsifier, and 2.0 g/mL for the others.

#### 2.5.2. Determination of Dispersibility of BP Emulsion

We added 99.5 mL of distilled water into a 100 mL graduated cylinder, and then dripped 0.5 mL of the emulsion through a pipette to observe its dispersion state. If it can disperse automatically in the form of clouds and mist and there are no visible particles, it is considered excellent. If it sinks in a flocculent shape, it is considered qualified [16], and slight oscillation was applied.

#### 2.5.3. Determination of Emulsification Stability of BP Emulsion

We determined emulsion stability according to the method in GB/T 1603-2001: Add 100 mL of 28~32 °C standard hard water into a 250 mL beaker. Use a pipette to absorb 1 mL of the emulsifiable oil sample (diluted by 100 times). Slowly add standard hard water under continuous stirring. Continue to stir at a speed of 2~3 r/s for 30 s after adding the emulsifiable oil. Immediately move the emulsion to a clean and dry 100 mL measuring cylinder. Place the measuring cylinder in a constant-temperature water bath and let it stand for 1 h within the range of 30 °C ± 2 °C to observe the separation of the emulsion. If there is no floating oil (cream), conduct precipitation in the measuring cylinder, and it is regarded as lotion if the stability is acceptable [17].

### 2.6. Determination of Germination of Weed Seeds by BP Emulsion

Take 6 fresh and plump weed seeds, including *C. album*, *E. benth*, *M. verticillate*, *A. retroflexus*, *L. holosteroides*, and *T. arvense*, and soak them in a 75% ethanol solution for 5 min for surface disinfection, and then thoroughly clean them with distilled water. Each treatment group consists of 3 replicates, with 20 seeds in each group. After soaking the seeds in a sterile PDB medium (CK) and 0.75% BP emulsion for 8 h, the seeds were transferred to a 9 cm culture dish with 2 layers of saturated filter paper. The dish was incubated in a constant-temperature control room at 28 °C, 80% humidity, and a light-to-dark ratio of 12:12 for 7 days. The dish was supplemented with an equal amount of sterile water daily using the weighing method.

Following Li Weiqing’s method [18], observe and record the germination of the first seed every day, and then calculate the germination rate, germination potential, and germination index after the experiment. When the length of the embryo exceeds half of the length of the seed, it is considered to have germinated. Measure the sprout length and root length of the seedlings on the 7th day by placing the seedlings flat to maintain their natural extension, measuring them three times with a ruler and taking the average, and calculating the seed vigor index.Germination rate (GR) = (number of germinated seeds/number of tested seeds) × 100%Germination index (GI) = ∑(Gt/Dt)Seed Vitality Index (VI) = GI × average sprout length of the 7th day seedlings

### 2.7. Determination of hCrop Seed Germination by BP Emulsion

Fresh and plump seeds of eight common crops in Qinghai Province, including *C. annuum*, *C. sativus*, *S. oleracea* L., *L. Asparagina*, *P. sativum*, *H*. *vulgare*, *T. aestivum*, and *B. campestris,* were taken using the same method as in the determination of germination of weed seeds by BP emulsion. After the experiment, the germination rate, germination potential, germination index, and seed vigor index were calculated.

### 2.8. Determination of Phytotoxicity of BP Emulsion on Potted Weeds

Eight major weeds in Qinghai Province, including *C. album*, *E. benth*, *M. verticillate*, *A. retroflex*, *T. arvense*, *G. spurium*, *L. holosteroides*, and *A. fatua*, were planted in pots (Φ = 15 cm) and cultured indoors. The 0.75% BP emulsifiable concentrates were inoculated into potted weed plants at the 4–7 leaf stage of normal growth by the spray inoculation method for three consecutive days, and the inoculation amount was 15 mL/pot. The inoculated weed plants were incubated in plastic bags for 24 h, and then placed in an artificial climate box at 25~30 °C, with L∥D = 12 h∥12 h. Each treatment was repeated three times, and 20 seeds were treated each time. The plants inoculated with sterile water were used as controls. After 7 days, the incidence of the inoculated weeds was observed, and the incidence and the fresh weight control effect were calculated [19].$$\mathrm{Disease}\;\mathrm{incience}=\frac{\;\mathrm{The}\;\mathrm{number}\;\mathrm{of}\;\mathrm{diseased}\;\mathrm{leaves}}{\mathrm{Investigate}\;\mathrm{the}\;\mathrm{total}\;\mathrm{number}\;\mathrm{of}\;\mathrm{leaves}}\times 100\%$$



$$\mathrm{Disease}\;\mathrm{index}=\frac{Grade\;of\;diseased\;leaves\times Corresponding\;level\;number}{Survey\;the\;total\;number\;of\;leaves\times highest\;level\;number}\times 100\%$$





$$\mathrm{Disease}\;\mathrm{index}=\frac{Control\;fresh\;weight-treatment\;fresh\;weight}{treatment\;fresh\;weight}\times 100\%$$



### 2.9. The Safety of BP Emulsion on Potted Crops

Eight common crops in Qinghai Province, including *C. annuum*, *C. sativus*, *S. oleracea*, *L. Asparagina*, *P. sativum*, *H.vulgare*, *T. aestivum*, and *B. campestris*, were planted in flower pots (Φ = 15 cm) and cultured indoors. We inoculated the 0.75% BP emulsion onto crop plants at the 4–7 leaf stage using the method mentioned above in the determination of phytotoxicity of the BP emulsion on potted weeds. Each treatment was repeated three times, and each treatment had 20 seeds. The crops inoculated with water were used as the control, and the disease situation was investigated 7 days later. The safety evaluation criteria for crops are as follows: NS indicates that the plant is asymptomatic (no disease spots and normal plant growth); LS indicates a slight impact (scattered disease spots on the leaves and slightly controlled growth and development); MS indicates moderate susceptibility (lesions appear on 1/5 to 1/4 of the leaf area, and growth is inhibited); SS represents severe disease (a large number of plants die, and their growth and development are severely controlled).

### 2.10. The Phytotoxicity of BP Emulsion on Weeds in Triticum aestivum Field Plots

(1)Experimental site information

The experimental site is located in Shenzhong Township, Huangyuan County, Qinghai Province, with a longitude of 101.19, a latitude of 36.73, and an altitude of 2475 m. The annual precipitation is 487 mm, and the annual average temperature is 3.0 °C. The organic matter content of the soil is 17.04 g/kg, with a pH of 8.18 and chestnut soil. The experimental site has a flat terrain and belongs to the Sichuan irrigation area, with an even distribution of weeds.

(2)Experimental design

A total of 6 treatments were designed for the experiment, including 1.00% BP emulsion 20 L/acre, 0.75% BP emulsion 20 L/acre, 0.50% BP emulsion 20 L/acre, 10% Tribenuron-methyl WP 2.5 g/acre (1/4 conventional dosage), 10% Tribenuron-methyl WP 10 g/acre (conventional dosage), and a blank control (Ck). Each treatment was repeated 3 times, with a total of 18 experimental plots. Each plot was 13.60 m long, 2.40 m wide, and had an area of 32.64 m^3^. The plots were randomly arranged.

(3)Application time and method

After *T. aestivum.* is sown, the pesticide is applied at the seedling stage, and the artificial backpack-type electric spray is used in each plot to spray the pesticide.

(4)Drug efficacy investigation

Three random samples were taken from each community, with a sample area of 0.25 m^2^. After 40 days of treatment, the number of remaining plants and the fresh weight of above-ground parts were investigated to calculate the plant efficiency and fresh weight efficiency. Plant control effectiveness (%) = (the number of weeds growing in the control area - the number of weeds growing in the treatment area)/the number of weeds growing in the control area × 100, while fresh weight control effect (%) = (the fresh weight of weed growth in the control area – the fresh weight of weed growth in the treatment area)/the fresh weight of weed growth in the control area × 100.

(5)Drug harm investigation

Conduct irregular observations to compare the response symptoms, seedling growth, leaf color, and later plant height of spring *Triticum aestivum* L. in different treatment areas by comparing the growth and development of crops in the control field, and comprehensively evaluate the growth and disease symptoms.

### 2.11. Data Statistical Analysis

The experimental data was analyzed and statistically analyzed using Excel 2019 and the SPSS 26.0data analysis software. Differences were assessed for significance using one-way analysis of variance (ANOVA) followed by the Least Significant Difference (LSD) test.

## 3. Results and Analysis

### 3.1. Selection of BP Emulsion Solvent

As shown in Figure 1 and Table 1, methanol showed the highest solubility for BP, followed by dimethyl sulfoxide and anhydrous ethanol, and xylene has the worst solubility. Considering the solubility and cost factors of six organic solvents, methanol is the optimal solvent for BP emulsion.

### 3.2. Formulation and Determination of BP Emulsifiable Concentrate

#### 3.2.1. Screening of Emulsifiers

As shown in Figure 2, using BP as the raw material to prepare the emulsion, six emulsifiers with concentrations of 1.5 g/mL and 2.0 g/mL were added to a solvent containing BP, resulting in a total of 12 emulsion formulations that basically meet the general properties of emulsion formulations and have a transparent appearance.

#### 3.2.2. Determination of Dispersibility and Stability of BP Emulsion

As shown in Table 2 in Figure 3, among the 12 emulsifiers produced by 6 emulsifiers, the emulsifiers produced by Nong Ru 1601, Tween-20, and Tween-80 at a concentration of 2. 0 g/mL have strong dispersibility and can appear cloudy or cotton-like when dropped into water. The dispersity of 12 kinds of emulsifiable concentrates is qualified after mixing, and they are a milky white lotion. After pouring them into a plate and letting them stand at 28 °C for 1 h, no upper oil slick and bottom sediment are found in Nong Ru 1601, Tween-20, and Tween-80 at the content of 2. 0 g/mL, as shown in Figure 4. Under the condition of a concentration of 2. 0 g/mL, the appearance, emulsion dispersibility, and stability of the Tween-20 emulsion fully meet the requirements of emulsion. Combining 2. 1 and 2. 2, it can be determined that the BP emulsion is composed of BP, methanol, and Tween-20 in a ratio of 1:1:2 g/mL.

#### 3.2.3. Determination of Germination of Weed Seeds by BP Emulsion

As shown in Figure 5 and Table 3, after adding the 0.75% BP emulsion to the weed seed culture medium for 7 days, all six weed seeds in the control group showed germination rates above 60%, while the six weed seeds in the treatment group did not germinate. This indicates that the 0.75% BP emulsion has inhibitory effects on the germination of six weed seeds, including *A*. *retroflexus*, *C*. *album*, *L. holosteoides*, *E. benth*, *T*. *arvense*, and *M*. *verticillate*. BP emulsion has weed control potential.

#### 3.2.4. Determination of Phytotoxicity of BP Emulsion on Potted Weeds

As shown in Figure 6 and Table 4, after spraying 0.75% BP emulsifiable concentrates for 7 days, *A*. *retroflexus*, *T*. *arvense*, and *L. holosteoides* have the best biological control effect, with the incidence and disease index reaching 100% and the control effect of fresh weight being 79.59%, 72.07%, and 73.87%, and their plants completely withered and died. The biological control effect of *G*. *spurium*, *C*. *album*, and *M*. *verticillate* was the second-best, with the incidence reaching 100% and the disease index reaching more than 90%. Some plants were not completely withered and died. The biological control effect of 0.75% BP emulsifiable concentrates on *E. benth* was the worst. The small leaves at the bottom of the large leaves were partially diseased and died. The incidence, disease index, and fresh weight of *E. benth* were 76.87%, 54.25%, and 25.05%. Based on the incidence, disease index, and fresh weight control effect, the 0.75% BP emulsifiable concentrate has extremely significant phytotoxicity on eight weeds, and from high to low, it is shown as *A*. *retroflexus* > *L. holosteoides* > *T*. *arvense* > *G*. *spurium* > *M*. *verticillate* > *C*. *album* > *A. fatua* > *E. benth*. The BP emulsifiable concentrate has the potential to develop into a new herbicide.

#### 3.2.5. Determination of Crop Seed Germination by BP Emulsion

As shown in Figure 7 and Table 5, after adding the 0.75% BP emulsion to the crop seed culture medium for 7 days, there was no significant difference in the germination rate, shoot length, root length, germination index, and vitality index of *B*. *campestris* seeds between the treatment group and the control group. The germination rate was 92%, indicating that the 0.75% BP emulsion is safe for *B*. *campestris* seeds. Compared with the control group, the germination length and vitality index of *T*. *aestivum* and *H*. *vulgare* seeds in the treatment group showed significant differences, while other indicators showed no significant differences. The germination rates were 93% and 89%, respectively, indicating that the 0.75% BP emulsion affects the growth of *T*. *aestivum* and *H*. *vulgare* seeds’ stems and leaves. Therefore, this emulsion may have an impact on nutrient supply during the growth process of *T*. *aestivum* and *H*. *vulgare*. The 0.75% BP emulsion has an inhibitory effect on the growth of *P*. *sativum* seeds, and the germination rate, shoot length, and root length of *P*. *sativum* crops in the treatment group were significantly lower than those in the control group. The 0.75% BP emulsion has a serious impact on the germination and growth of seeds in four crops: *C. sativus*, *L. asparagina*, *S. oleracea*, and *C. annuum*. The treatment group showed significant differences in the germination rate, shoot length, root length, germination index, and vitality index, with the germination rate below 20%. In summary, based on the comprehensive determination of the seed germination rate, sprout length, root length, germination index, and vitality index, the 0.75% BP emulsion is relatively safe for *B*. *campestris*, *T*. *aestivum*, and *H*. *vulgare* seeds, and it can be applied during the field planting period of the *B*. *campestris*, *T*. *aestivum*, and *H*. *vulgare* crops.

#### 3.2.6. The Safety of BP Emulsion on Potted Crops

As shown in Figure 8 and Table 6, after spraying the 0.75% BP emulsion for 7 days, it showed no phytotoxicity to *P*. *sativum* and *T*. *aestivum*. Compared with the control, there was no significant change in crop growth and plant height, indicating no response (NS). BP emulsion can be used as a new environmentally friendly microbial herbicide for the *P*. *sativum* and *T*. *aestivum* crops in the field. Both *C*. *annuum* and *H*. *vulgare* showed mild susceptibility (LS), while *H*. *vulgare* showed mild height suppression and slight yellowing of leaf tips. *C*. *annuum* showed black disease spots on its leaves, but after later observation, they had no effect on crops. They 0.75% BP emulsifiable concentrates have the effect of inhibiting plant height of *B*. *campestris* and *L. asparagina*, and some plant leaves are diseased and dead, with incidences of 48% and 60%, respectively, showing moderate susceptibility (MS). The 0.75% BP emulsifiable concentrates have high phytotoxicity to *C. sativus* and *S. oleracea*. Compared with the control, it shows plant height inhibition, as well as withered and yellow leaves, and some plant leaves are diseased and dead. Its incidences are 85% and 82%, respectively, indicating severe susceptibility (SS).

#### 3.2.7. The Pathogenic Effect of BP Emulsion on Weeds in *Triticum aestivum* Field Plots

As shown in Figure 9 and Table 7, after the drugs with different concentrations of 10% Tribenuron-methyl WP and BP emulsifiable concentrates are evenly sprayed in each plot, the control effects of the 1.00%, 0.75%, and 0.50% BP emulsifiable concentrates on weed plants in *T. aestivum* fields are 94.25%, 81.73%, and 0%, respectively, and the control effects of fresh weight are 87.12%, 59.47%, and 21.65%. The phytotoxicity of field weeds such as *C*. *album*, *E. benth*, and *A*. *retroflexus* mainly manifests as a large number of withered and dead leaves, severe growth inhibition, and more significant phytotoxicity as the dilution ratio of the BP emulsion decreases. The plant control effects of 10% Tribenuron-methyl WP 10 g/acre and 2.5 g/acre were 99.37% and 97.56%, respectively, and the fresh weight control effects were 87.15% and 76.48%, respectively, which were more significant compared to the BP emulsion. Compared with the Ck control plot, the *T. aestivum* plants in the 1.00% BP emulsion treatment plot showed wilting symptoms at the leaf tip, while the *T. aestivum* plants in the 0.75% emulsion treatment plot did not show any disease spots or growth symptoms. The *T. aestivum* plants in the 10% Tribenuron-methyl WP treatment plot did not show any disease spots or growth symptoms. In summary, the BP emulsion has a weed control effect on weeds such as *C*. *album*, *E. benth*, and *A*. *retroflexus* in *T. aestivum* fields at concentrations ranging from 1.00% to 0.75%. However, the effect is slightly worse than that of the 10% Tribenuron-methyl WP chemical herbicide, and neither of them showed any symptoms of damage to the growth of *T. aestivum* crops in the field.

## 4. Discussion

Microbial-derived herbicides have a unique chemical structure and are easily degradable. Compared to chemical herbicides, they have unique advantages in weed resistance, pesticide residue, environmental pollution, and other aspects [20,21]. Due to the above reasons, more and more researchers are joining the research ranks of microbial herbicides. Microbial herbicides usually need to be processed into suitable dosage forms for storage and use, such as powders, emulsions, pastes, etc. [22]. This study used the metabolite BP from the fungus HY-02 as the raw material and followed the method in GB/T 1603-2001 to screen solvents and emulsifiers to develop an emulsion-type microbial herbicide. Through experiments, the most suitable solvent is methanol, which can dissolve BP (1 g/mL), and the most suitable emulsifier is Tween-20 at a content of 1.5 g/mL. Under these conditions, the appearance, emulsion dispersibility, and stability of the BP emulsion fully meet the requirements of emulsion.

Developing and utilizing microbial resources as biological herbicides is one of the important objectives of microbial research [23]. Numerous studies have shown that secondary metabolites and their derivatives produced by plant pathogens can be used to control weeds and develop into biological herbicides [24], such as the highly toxic *macul-sion* produced by *Alternaria altemate*, which is highly toxic to *Centaurea aculosa* [25]; *cypein* produced by *Fusarium oxysporum*; and toxins such as *cypein* produced by *Ascochyta* and *Phoma* fungi, as well as *tenuazonic* acid (TeA), AAL, ACT, AM, and other toxins produced by *Alternaria* fungi [26]. Determining the scope of use is an important step in developing efficient biological herbicides [27]. The main criteria for herbicides are phytotoxicity to a wide range of weeds and safety to crops [28]. This study used the 0.75% BP emulsion for seed germination determination. After adding the BP emulsion to the weed seed culture medium, the BP emulsion had an inhibitory effect on the germination of six weed seeds. After adding the crop seed culture medium, the BP emulsion is safe for *B*. *campestris* seeds. It has an inhibitory effect on the growth of stems and leaves of cereal seeds (*T*. *aestivum*, *H*. *vulgare*). It has a significant impact on *C. sativus*, *L. asparagina*, *S. oleracea*, and *C. annuum* seeds, with germination rates all below 20%. When 0.75% BP emulsion concentrates were sprayed on the living potted plants of eight weeds, it was found that the incidence of BP emulsion concentrates on *A. retroflexus*, *L. holosteoides*, *T*. *arvense*, *G*. *spurium*, *M*. *verticillate*, *C*. *album*, and *A. fatua* reached 100%. According to the crop safety evaluation, BP emulsion concentrates have no disease (NS) on *P*. *sativum* and *T*. *aestivum*, slight plant height inhibition and slight susceptibility (LS) on highland *H*. *vulgare* and peppers, high phytotoxicity on *C. sativus* and *S. oleracea*, and some plant leaves are diseased and dead, with incidences of 85% and 82%, respectively.

BP has the advantages of low toxicity, good performance, and rapid hydrolysis, and it is one of the commonly used preservatives. It is frequently used in the food industry of many countries in Asia and Europe and is also a key food preservative developed in China [29]. BP is widely used in cosmetics, personal care products, daily necessities, the pharmaceutical industry, and other fields due to its excellent antibacterial effect [30,31]. In addition, this study found that BP can also be widely used in the field of herbicides. In this experiment, BP emulsion concentrates were diluted with different concentrations of spray and inoculated into *T*. *aestivum* fields. The application was carried out at the seedling stage. It was found that BP emulsion concentrates had a herbicidal effect on weeds such as *C*. *album*, *E. benth*, and *A*. *retroflexus* at the concentration of 1.00%~0.75%, and they had high safety on *T*. *aestivum* crops in the field. At the same time, this experiment compared the weed control effects of the BP emulsion and 10% Tribenuron-methyl WP on *T. aestivum* fields. It was found that 10% Tribenuron-methyl WP had slightly better weed control effects on *T. aestivum* fields than the BP emulsion, but it did not widen the gap. With the development of the BP emulsion, the difference in weed control effects between the BP emulsion and chemical herbicides will become smaller and smaller. Moreover, the low toxicity and rapid hydrolysis of the BP emulsion also make it have broad application potential.

Herbicides have become an integral component of agricultural production, with over 4.1 million tons of pesticides used globally each year, approximately 60% of which are chemical herbicides [32]. A significant proportion of these herbicides enters the environment, leading to adverse effects on non-target biological communities [33]. Pyroxasulfone, s-metolachlor, and dimethenamid-p exhibit persistent environmental impacts, remaining in biological systems and soil for months to years [34]. Their interaction with soil organic matter can negatively affect subsequent crop rotations [35]. Furthermore, pyroxasulfone herbicides have been shown to pose considerable toxicity risks to both humans and wildlife [36]. In contrast, BP offers a favorable environmental safety profile due to its diverse degradation pathways, including advanced oxidation processes, biodegradation, and photocatalysis [37]. BP is rapidly hydrolyzed in the human body and excreted via urine [38]. In soil, it is degraded through chemical reactions, adsorption, and microbial metabolism, with the latter being the predominant route [39]. Studies have demonstrated that BP exhibits low persistence in soil, with over 90% degradation occurring within three days at concentrations ranging from 0 to 1000 mg/kg [40]. In aquatic environments, BP can be metabolized by microalgae and other microorganisms as a carbon source, and it also undergoes abiotic hydrolysis [41]. Additionally, under simulated solar irradiation in the presence of sensitizers such as rose bengal and aluminum chloride phthalocyanine tetrasulfonate, BP undergoes rapid photodegradation [42]. In summary, BP shows considerable potential for application in the herbicide sector. Its strong antibacterial properties, low toxicity, and high degradability enable it to provide dual benefits of weed control and antimicrobial activity, thereby supporting the diversification and advancement of herbicide technologies.

## 5. Conclusions

This article compares the effects of different emulsifiers, solvents, dispersibility, and stability on the preparation of the BP emulsion and ultimately determines that the BP emulsion is a mixture of BP, methanol, and Tween-20 in a ratio of 1:1:2 g/mL. Basic research was conducted on the BP emulsion through seed germination inhibition, phytotoxicity of living plants, crop safety, and field effects. The results showed that the BP emulsion can be developed into a field microbial herbicide for Poaceae (*T*. *aestivum* and *H*. *vulgare*) crops. In subsequent experiments, the proportion of the BP emulsion will be adjusted to enhance the effect, and it will be compounded with common chemical herbicides and other agricultural reagents to reduce the application of chemical herbicides while also conducting in-depth research on the diversification of herbicide prospects. A new environmentally friendly microbial herbicide that can be applied in multiple fields of agricultural production will be developed.

## Figures and Tables

**Figure 1 plants-14-03041-f001:**
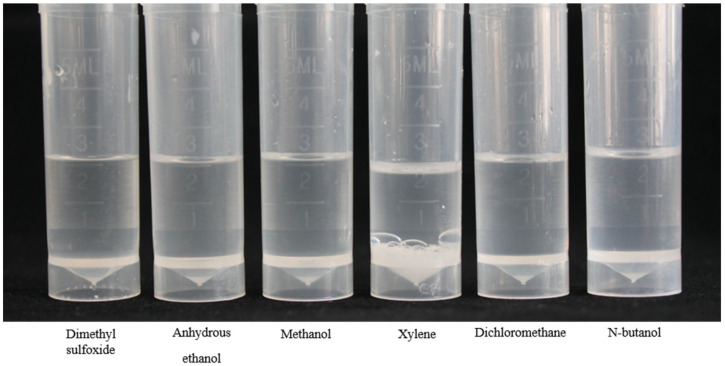
Determination of solubility of BP in 6 different solvents.

**Figure 2 plants-14-03041-f002:**
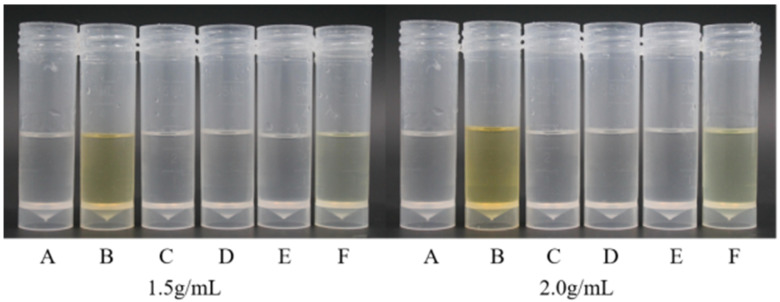
The state of 12 emulsifiers prepared into emulsion. Note: A, Tween-20; B, Tween-80; C, Nong Ru 1601; D, OV-5; E, Fatty alcohol polyoxyethylene ether phosphate ester; and F, OF-4816-B.

**Figure 3 plants-14-03041-f003:**
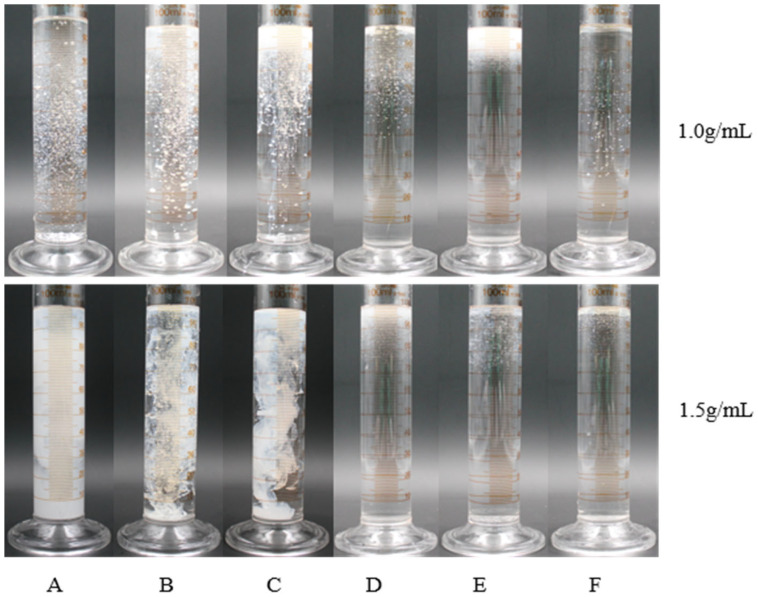
Dispersion effect of different types of emulsion. Note: A, Tween-20; B, Tween-80; C, Nong Ru 1601; D, OV-5; E, Fatty alcohol polyoxyethylene ether phosphate ester; and F, OF-4816-B.

**Figure 4 plants-14-03041-f004:**
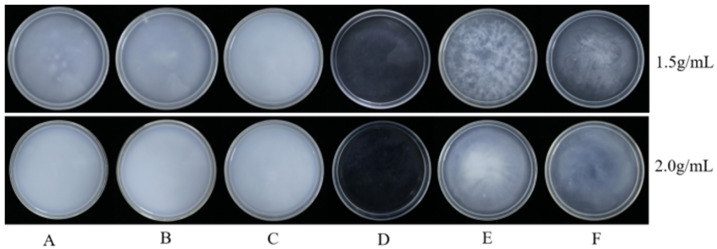
Stability effect diagram of different types of emulsions. Note: A, Tween-20; B, Tween-80; C, Nong Ru 1601; D, OV-5; E, Fatty alcohol polyoxyethylene ether phosphate ester; and F, OF-4816 B.

**Figure 5 plants-14-03041-f005:**
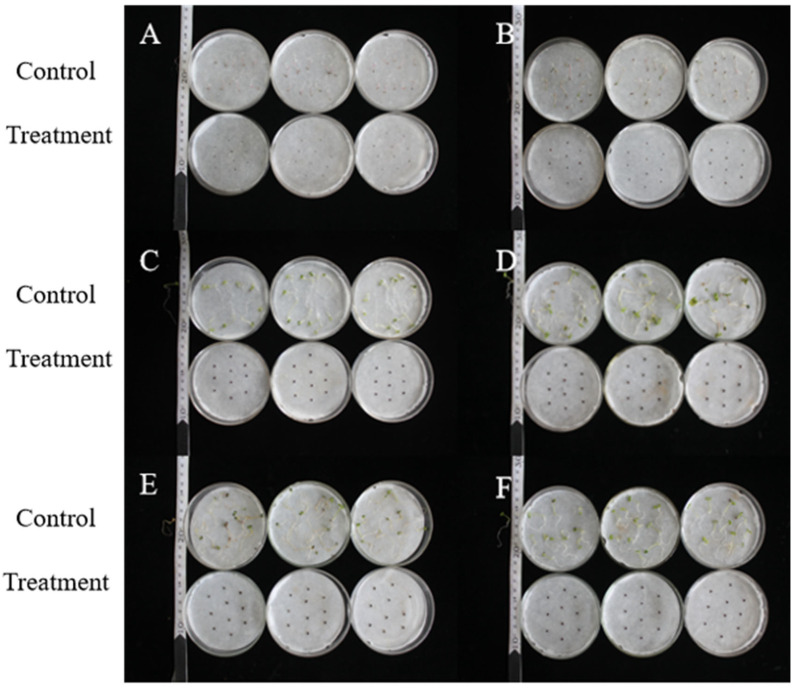
Determination of 0.75% BP emulsion on weed seed germination. Note: A, *A. retroflexus*; B, *C. album*; C, *L*. *holosteoides*; D, *E. benth;* E, *T. arvense*; and F, *M. verticillata*.

**Figure 6 plants-14-03041-f006:**
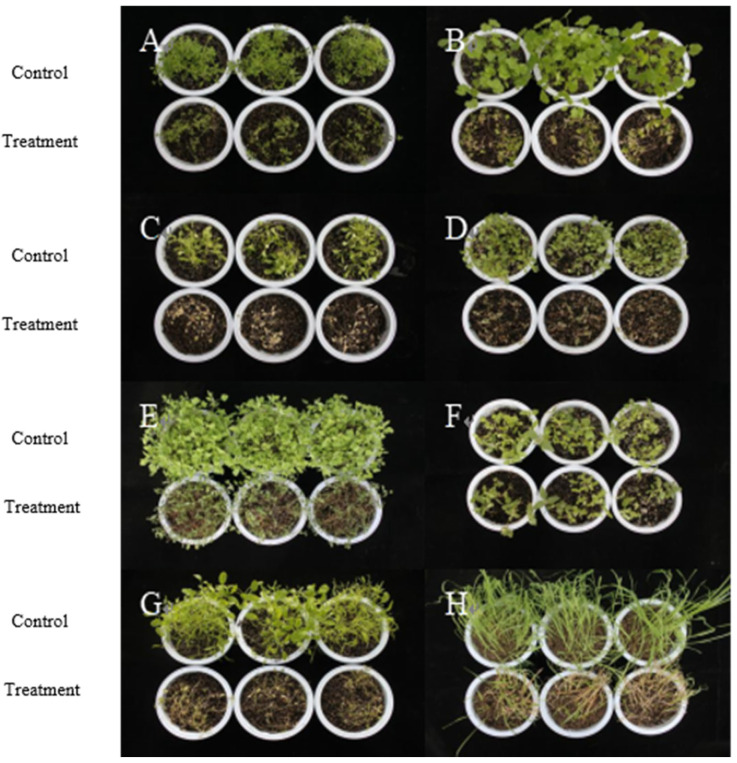
Phytotoxicity of 0.75% BP emulsion on potted weeds. Note: A, *G. spurium*; B, *M. verticillate*; C, *T. arvense*; D, *A. retroflexus*; E, *C. album*; F, *E. benth;* G, *L. holosteoides*; and H, *A. fatua*.

**Figure 7 plants-14-03041-f007:**
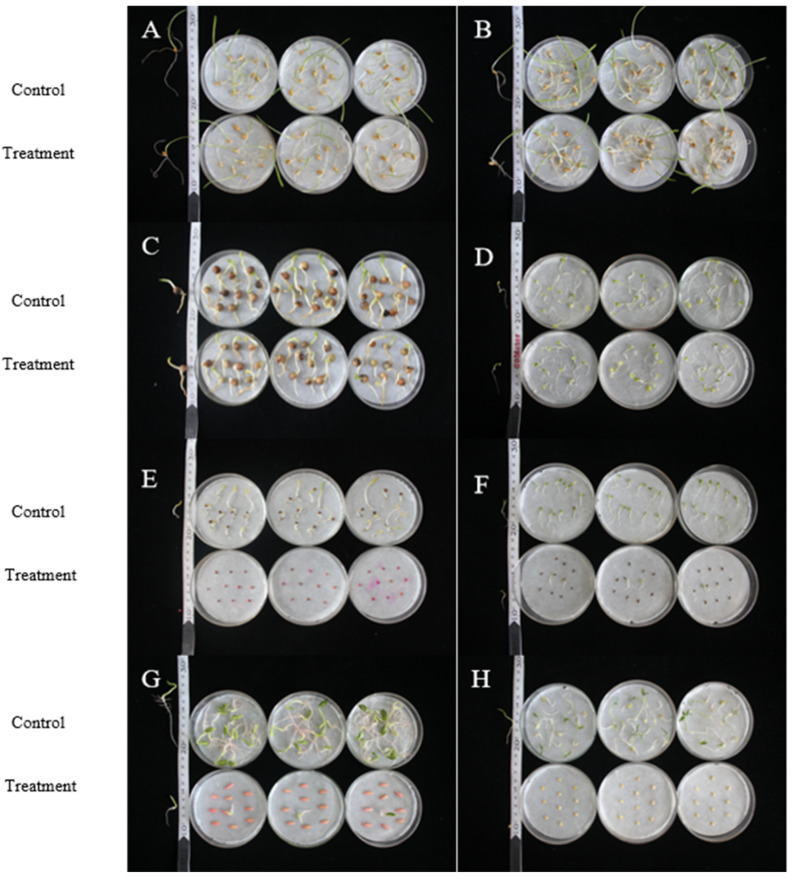
Determination of crop seed germination by 0.75% BP emulsion. Note: A, *T. aestivum*; B, *H. vulgare*; C, *P. sativum*; D, *B. campestris*; E, *S. oleracea*; F, *L. asparagina*; G, *C. sativus*; and H, *C. annuum*.

**Figure 8 plants-14-03041-f008:**
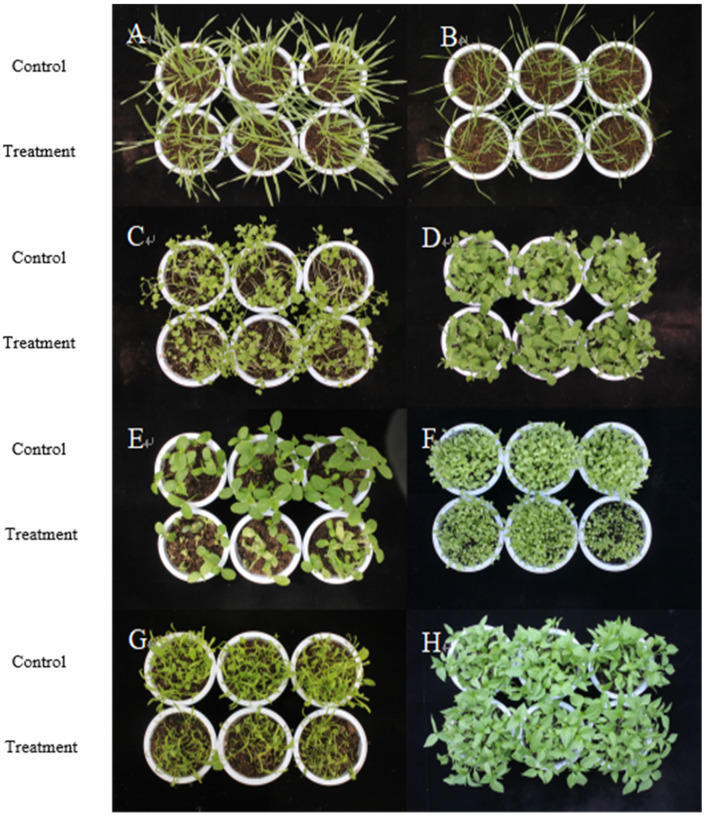
Safety of BP emulsion on crops. Note: A, *H. vulgare*; B, *T. aestivum*; C, *B. campestris*; D, *P*. *sativum*; E, *C. sativus*; F, *L. asparagina*; G, *S. oleracea*; and H, *C. annuum*.

**Figure 9 plants-14-03041-f009:**
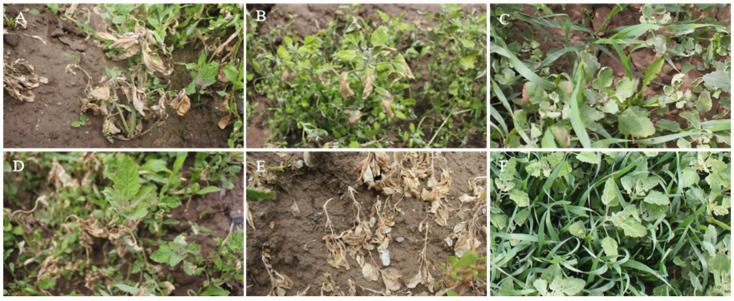
The effect of different concentrations of BP emulsion on weed control in *T*. *aestivum* fields. Note: (**A**), 1.00% BP emulsion 20 L/acre; (**B**), 0.75% BP emulsion 20 L/acre; (**C**), 0.50% BP emulsion 20 L/acre; (**D**), 10% Tribenuron-methyl WP 2.5 g/acre; (**E**), 10% Tribenuron-methyl WP 10 g/acre; and (**F**), Ck.

**Table 1 plants-14-03041-t001:** Determination of solubility of BP in 6 different solvents.

Solvent Type	Solute/Solvent//g/mL
Dimethyl sulfoxide	1/2
Anhydrous ethanol	1/2
Methanol	1/1
Xylene	1/10
Dichloromethane	1/3
N-butanol	1/6

**Table 2 plants-14-03041-t002:** Results of BP emulsion dispersibility and stability test.

Serial Number	Emulsifier	Usage (g)	Appearance of Emulsion	Dispersibility and Stability of Emulsion		
				Initial Infusion	After Stirring	1 h Later
1	Tween-20	1.5	White transparent	Droplet-like sinking and spreading of oil	Milky white lotion	unqualified
		2.0	White transparent	Cloudy dispersion	Milky white lotion	excellent
2	Tween-80	1.5	Yellow transparent	Droplet-like sinking and spreading of oil	Milky white lotion	unqualified
		2.0	Yellow transparent	Cotton-like sinking and spreading	Milky white lotion	excellent
3	Nong Ru 1601	1.5	White transparent	Droplet-like sinking and spreading of oil	Milky white lotion	unqualified
		2.0	White transparent	Cloudy dispersion	Milky white lotion	excellent
4	OV-5	1.5	White transparent	Droplet-like sinking and spreading of oil	Milky white lotion	unqualified
		2.0	White transparent	Droplet-like sinking and spreading of oil	Milky white lotion	unqualified
5	Fatty alcohol polyoxyethylene ether phosphate ester	1.5	White transparent	Droplet-like sinking and spreading of oil	Milky white lotion	unqualified
		2.0	White transparent	Droplet-like sinking and spreading of oil	Milky white lotion	unqualified
6	OF-4816-B	1.5	Light yellow transparent	Droplet-like sinking and spreading of oil	Milky white lotion	unqualified
		2.0	Light yellow transparent	Droplet-like sinking and spreading of oil	Milky white lotion	unqualified

Note: The raw materials for sequences 1–6 are BP, and the solvent is methanol.

**Table 3 plants-14-03041-t003:** Determination of 0.75% BP emulsion on weed seed germination.

Weeds Species	Germination Rate	Root Length	Bud Growth	Germination Index	Vitality Index
*T*. *arvense*	0 ^b^	0 ^b^	0 ^b^	0 ^b^	0 ^b^
CK	86 ^a^	4.32 ^a^	2.56 ^a^	6.15 ^a^	15.74 ^a^
*M. verticillata*	0 ^b^	0 ^b^	0 ^b^	0 ^b^	0 ^b^
CK	67 ^a^	4.83 ^a^	1.94 ^a^	4.45 ^a^	8.63 ^a^
*E. benth*	0 ^b^	0 ^b^	0 ^b^	0 ^b^	0 ^b^
CK	7 ^a^	4.16 ^a^	2.28 ^a^	5.33 ^a^	12.15 ^a^
*C. album*	0 ^b^	0 ^b^	0 ^b^	0 ^b^	0 ^b^
CK	83 ^a^	2.28 ^a^	0.94 ^a^	5.85 ^a^	5.45 ^a^
*L. holosteoides*	0 ^b^	0 ^b^	0 ^b^	0 ^b^	0 ^b^
CK	77 ^a^	4.76 ^a^	1.87 ^a^	5.35 ^a^	10.01 ^a^
*A. retroflexus*	0 ^b^	0 ^b^	0 ^b^	0 ^b^	0 ^b^
CK	83 ^a^	1.65 ^a^	1.13 ^a^	5.82 ^a^	6.57 ^a^

Note: Lowercase letters indicate whether there is a significant difference in data between the control group and the experimental group of the same weed (*p* < 0.05).

**Table 4 plants-14-03041-t004:** Phytotoxicity of 0.75% BP emulsion on different weeds.

Weeds Species	Disease Incidence	Disease Index	Disease Index
*M. verticillata*	100 ^a^	98.89 ± 1.11 ^a^	71.79 ± 2.15 ^a^
*T*. *arvense*	100 ^a^	100 ^a^	72.07 ± 1.56 ^a^
*E. benth*	76.87 ± 1.32 ^b^	54.25 ± 2.98 ^c^	25.05 ± 2.61 ^c^
*C. album*	100 ^a^	91.33 ± 3.12 ^a^	79.41 ± 2.23 ^a^
*G*. *aparine*	100 ^a^	96.67 ± 2.16 ^b^	64.24 ± 1.62 ^b^
*A. retroflexus*	100 ^a^	100 ^a^	79.59 ± 1.58 ^a^
*Lepyrodiclis holosteoides*	100 ^a^	100 ^a^	73.87 ± 1.94 ^a^
*A. fatua L.*	100 ^a^	75.00 ± 2.37 ^b^	79.87 ± 1.82 ^a^

Note: Lowercase letters indicate significant differences between data in the same column for different crops (*p* < 0.05).

**Table 5 plants-14-03041-t005:** Determination of crop seed germination by 0.75% BP emulsion.

Weeds Species	Germination Rate	Root Length (cm)	Bud Growth (cm)	Germination Index	Vitality Index
*T*. *aestivum*	93 ^a^	6.53 ^a^	4.43 ^b^	7.23 ^a^	32.03 ^b^
CK	98 ^a^	6.93 ^a^	6.11 ^a^	7.73 ^a^	47.23 ^a^
*H*. *vulgare*	89 ^a^	4.93 ^a^	4.86 ^b^	6.70 ^a^	32.56 ^b^
CK	97 ^a^	5.26 ^a^	6.68 ^a^	7.68 ^a^	51.30 ^a^
*P*. *sativum*	58 ^b^	2.47 ^a^	1.91 ^b^	3.15 ^b^	6.02 ^b^
CK	90 ^a^	2.94 ^a^	2.56 ^a^	6.43 ^a^	16.46 ^a^
*B*. *campestris*	92 ^a^	3.51 ^a^	1.36 ^a^	7.56 ^a^	10.28 ^a^
CK	95 ^a^	3.68 ^a^	1.62 ^a^	7.90 ^a^	12.79 ^a^
*C*. *sativus*	20 ^b^	2.66 ^b^	1.71 ^b^	1.62 ^b^	2.77 ^b^
CK	95 ^a^	5.93 ^a^	2.82 ^a^	7.26 ^a^	20.47 ^a^
*L. asparagina*	10 ^b^	0.63 ^b^	1.21 ^b^	0.75 ^b^	0.91 ^b^
CK	90 ^a^	2.17 ^a^	2.81 ^a^	7.61 ^a^	21.38 ^a^
*S*. *oleracea*	0 ^b^	0 ^b^	0 ^b^	0 ^b^	0 ^b^
CK	90 ^a^	0.63 ^a^	2.17 ^a^	7.64 ^a^	16.58 ^a^
*C*. *annuum*	10 ^b^	0 ^b^	0 ^b^	0 ^b^	0 ^b^
CK	90 ^a^	2.27 ^a^	3.65 ^a^	6.51 ^a^	23.76 ^a^

Note: Lowercase letters indicate whether there is a significant difference in data between the control group and the experimental group of the same crop (*p* < 0.05).

**Table 6 plants-14-03041-t006:** Safety of BP emulsion on crops.

Crops Tested	Inhibition Rate of Height	Disease Incidence	Safety Grade
*T*. *aestivum*	0	0	NS
*B*. *campestris*	1	48	MS
*P*. *sativum*	0	0	NS
*H*. *vulgare*	1	12	LS
*L. asparagina*	10	60	MS
*S*. *oleracea*	40	82	SS
*C*. *annuum*	0	32	LS
*C*. *sativus*	60	85	SS

**Table 7 plants-14-03041-t007:** Control effect of BP emulsion at different concentrations on weeds in *T*. *aestivum* fields after 40 days of treatment (%).

Treatment Group	Plant Preventing Effect%	Fresh Weight Efficiency%
1.00% BP emulsion 20 L/acre	94.25 ^b^	57.12 ^c^
0.75% BP emulsion 20 L/acre	81.73 ^c^	39.47 ^d^
0.50% BP emulsion 20 L/acre	0 ^d^	21.65 ^e^
10% Tribenuron-methyl WP 2.5 g/acre	97.56 ^ab^	76.48 ^b^
10% Tribenuron-methyl WP 10 g/acre	99.37 ^a^	87.15 ^a^
Ck	0 ^d^	0 ^f^

Note: Lowercase letters indicate significant differences (*p* < 0.05) between data in the same column.

## Data Availability

Checked and verified. No issues were found.

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
