# Peer review of "Preparation and Herbicidal Evaluation of Butyl Hydroxybenzoate Emulsion"

_plants, 2025, doi:10.3390/plants14193041_

Round 1
Reviewer 1 Report
Comments and Suggestions for Authors
The subject of the article is of considerable interest, particularly in light of the paucity of novel and efficacious herbicides. This issue is especially salient in the context of crop production. The development of mycoherbicides is not a novel concept; nevertheless, it persists in representing a considerable scientific challenge. The meticulous methodology and the precise execution of all experimental components are commendable.
A particularly intriguing issue is the effect of butyl paraben (BP) on weed germination: are the seeds truly non-viable, or merely in a state of dormancy? This distinction is critical for evaluating the efficacy of alternative strategies. In conclusion, the article is regarded as being of significant value and as a promising contribution to the field of sustainable weed management.
Specific comments:
Abstract
- Provide full scientific names of plant species.
- Introduction
- Add information on the technology and methods used to obtain butyl paraben (BP).
- Materials and Methods
2.2 Test weeds
- Present scientific names of weed species in italics, e.g., Chenopodium album
2.3 Test crops - Present full scientific names of crops in italics, e.g., Capsicum annuum
- Line 93: correct “Hordeumbulgare” → Hordeum vulgare; “Brassicapus” → Brassica campestris L.
2.5.8 The pathogenic effect of BP emulsion on weeds in Triticum aestivum L. field plots - Italicize scientific names in the subsection title (Triticum aestivum).
- Add details on soil and climatic conditions of the field experiment, as well as GPS coordinates.
- Results and Analysis
- Line 256: Ensure scientific names of weeds are italicized.
- Lines 277–278: Ensure scientific names of weeds are italicized.
- Lines 305–306: Ensure scientific names of weeds are italicized.
- Lines 327–328: Ensure scientific names of crops are italicized.
- Conclusion and Discussion
- Expand the discussion to include information on BP degradation processes in soil.
- Clarify the potential environmental risks associated with BP.
- Line 413: Present scientific names of plant families in italics.
English requires careful professional editing to improve clarity, consistency of terminology, and grammar. Please standardize Latin binomials in italics, define all abbreviations, and use “phytotoxicity” rather than “pathogenicity” when referring to herbicidal effects.
Author Response
Dear Dr. Reviewer,
Thank you very much for your letter and we are very grateful for the constructive comments and valuable recommendations given by you. Based on your suggestions and edits, we have carefully revised the manuscript.
Here below is our Point-by-point responses to the comments.
Comments and Suggestions
1*. A particularly intriguing issue is the effect of butyl paraben (BP) on weed germination: are the seeds truly non-viable, or merely in a state of dormancy?
Response1*. Thank you for the expert's question. Regarding the question of whether seeds are dormant or truly unable to survive, based on our existing data, we can only show that it can have an impact on seed germination rate and the growth of roots, stems, and shoots after germination. Without specific experimental data, we cannot determine whether it is dormant or unable to survive. However, we also appreciate the expert's guidance and will design an experiment to address this issue.
2*. Abstract, Provide full scientific names of plant species.
Response2*. The complete scientific names of the plant species in the abstract have been supplemented.
3*. 1.Introduction, Add information on the technology and methods used to obtain butyl paraben (BP).
1.Materials and Methods
2.2 Test weeds
Present scientific names of weed species in italics, e.g., Chenopodium album
Response3*. The weeds and crops names in 2.2 have been italicized.
4*. 2.3 Test crops
Present full scientific names of crops in italics, e.g., Capsicum annuum
Line 93: correct “Hordeumbulgare” → Hordeum vulgare; “Brassicapus” → Brassica campestris L.
Response4*. The italicization proposed in 2.3 has been carried out.
5*. 2.5.8 The pathogenic effect of BP emulsion on weeds in Triticum aestivum L. field plots
Italicize scientific names in the subsection title (Triticum aestivum).
Add details on soil and climatic conditions of the field experiment, as well as GPS coordinates.
Response5*. The italicized issues in the Materials and Methods section have been revised. Meanwhile, specific information about the experimental site has been added to the field experiment part of the Materials and Methods section.
6*. 3.Results and Analysis
Line 256: Ensure scientific names of weeds are italicized.
Lines 277–278: Ensure scientific names of weeds are italicized.
Lines 305–306: Ensure scientific names of weeds are italicized.
Lines 327–328: Ensure scientific names of crops are italicized.
Response6*. The italicization proposed in Results and Analysis has been carried out.
7*.4.Conclusion and Discussion
Expand the discussion to include information on BP degradation processes in soil.
Clarify the potential environmental risks associated with BP.
Line 413: Present scientific names of plant families in italics.
Response7*. In the discussion section, additional information was provided regarding the degradation of PB in the soil, and some hazards of chemical herbicides were also included. The italicization proposed in Line 413 has been carried out.
8*. Comments on the Quality of English Language
English requires careful professional editing to improve clarity, consistency of terminology, and grammar. Please standardize Latin binomials in italics, define all abbreviations, and use “phytotoxicity” rather than “pathogenicity” when referring to herbicidal effects.
Response8*. We revised the italicized issues in the text, checked the accuracy and consistency of the language, and changed "phytotoxicity" to "pathogenicity".
Reviewer 2 Report
Comments and Suggestions for Authors
The manuscript titled “ Preparation and Herbicidal Evaluation of Butyl Hydroxybenzoate Emulsion” presents the designing of microbial herbicide in form of emulsion, metabolite butyl hydroxybenzoate (BP) of the Alternaria in mixture with methanol, and Tween-20.
The paper is concise, well-written, and provides novel data on the development of a new environmentally friendly microbial herbicide for field application in grasses, characterized by antibacterial activity, low toxicity, and hydrolysis effects, which may promote diversification in herbicide development.
I have a few remarks:
- Did the authors test the effects of the individual components of the emulsion separately (BP, methanol, Tween-20) to rule out their independent contribution to the overall efficacy?
- Authors should separate the Discussion section from the Conclusion section.
- The Discussion section should be strengthened to better justify the use of the eco-friendly designed product compared to conventional weed control approaches.
Author Response
Dear Dr. Reviewer,
Thank you very much for your letter and we are very grateful for the constructive comments and valuable recommendations given by you. Based on your suggestions and edits, we have carefully revised the manuscript.
Here below is our Point-by-point responses to the comments.
Comments and Suggestions
1*. Did the authors test the effects of the individual components of the emulsion separately (BP, methanol, Tween-20) to rule out their independent contribution to the overall efficacy?
Response1*. During the experiment, we conducted a preliminary assessment of the emulsions under different emulsifiers and conditions. We found that the effect of other emulsifiers was inferior to that of 20. We initially hypothesized that 20 increased the permeability of the leaves. In the current experiment we are conducting, we are further expanding the screening range of the solvents and emulsifiers for this emulsion and conducting targeted experiments. Here, we would like to thank the reviewers for their valuable suggestions. We will incorporate this point you mentioned into the experiment as well.
2*. Authors should separate the Discussion section from the Conclusion section.
Response2*. The conclusion section and the discussion have been separated as per the suggestions.
3*. The Discussion section should be strengthened to better justify the use of the eco-friendly designed product compared to conventional weed control approaches.
Response3*. In the discussion section, we included relevant information about traditional chemical herbicides and conducted a comparison.
Reviewer 3 Report
Comments and Suggestions for Authors
The paper entitled “Preparation and Herbicidal Evaluation of Butyl Hydroxybenzoate Emulsion” describes the development of a new environmentally friendly microbial herbicide for weed control. The object of the current study is the metabolite butyl hydroxybenzoate (BP) of the HY-02 strain of Alternaria. The paper is accurately written, actual, presents new potent microbial herbicide with seed germination inhibition effect, pathogenicity of living plants, crop safety.The topic of Ms is original and possess novelty since the search of new environmentally friendly microbial herbicide for field application in grasses (T. aestivum, H. vulgare) crops is needed. The main question addressed by this research is the development of Butyl Hydroxybenzoate Emulsion as an excellent antibacterial, low toxic with rapid hydrolysis microbial herbicide for weed control.The conclusions consistent with the evidence and are detailed, the necessary arguments are presented and addressed to the main question posed. The figures and tables are concisely and clearly reflect the details of the discussed topic. The paper could be accepted to Plants after some corrections:
1) please change the Sections ito the following: Results and Discussion, Conclusions.
2) the choice of MeOH as a BP emulsion solvent seems not attractive in an industial scale due to its harmful and poisonous proterties.
3) please add a legend to Tables 3, 4, 7 (a and b meaning).
4) please name a herbicide used as a contol?
5) in Conclusion please formulate the advantages of BP in comparison with other herbicides.
Author Response
Dear Dr. Reviewer,
Thank you very much for your letter and we are very grateful for the constructive comments and valuable recommendations given by you. Based on your suggestions and edits, we have carefully revised the manuscript.
Here below is our Point-by-point responses to the comments.
Comments and Suggestions
1*. please change the Sections ito the following: Results and Discussion, Conclusions.
Response1*. The conclusion section and the discussion have been separated as per the suggestions.
2*. the choice of MeOH as a BP emulsion solvent seems not attractive in an industial scale due to its harmful and poisonous proterties.
Response2*. When choosing the solvent, we took this into account. However, considering the dissolution rate and the fact that adding water during use would reduce the toxicity to a safe level, we chose O. In the current experiment we are conducting, we are further expanding the screening range of the solvents and emulsifiers for this emulsion and conducting targeted experiments. Here, we would like to thank the reviewers for their valuable suggestions. We will incorporate this point you mentioned into the experiment as well.
3*. please add a legend to Tables 3, 4, 7 (a and b meaning).
Response3*. According to the suggestions, the legends of Tables 3, 4 and 7 have been modified.
4*. please name a herbicide used as a contol?
Response4*. First of all, I would like to express my sincere gratitude to this expert for the question. In our field experiments, we did use chemical herbicides as a control, but since the efficacy of our herbicide still had a certain gap compared to chemical herbicides, we did not include the chemical herbicide control at the beginning. However, after receiving guidance from experts, we believe it is necessary to include it in the experiment for a comparison. Although the herbicide we developed still has a certain gap compared to chemical herbicides, we also hope to strive to reach this goal in subsequent experiments. Finally, we would like to express our gratitude once again to the experts for their guidance.
5*. in Conclusion please formulate the advantages of BP in comparison with other herbicides.
Response5*. In the discussion section, we included relevant information about traditional chemical herbicides and conducted a comparison.
Round 2
Reviewer 1 Report
Comments and Suggestions for Authors
It's good to see that you appreciate all the comments.
Comments have been carefully addressed.
Minor comments:
Abstract:
- Line 21: edit „Cucumis sativus L., Lactuca sativa L. var. asparagina,“
- Line 36: edit „Hordeum vulgare L.)“
2.2 Test weeds
- Line 96: edit „Elsholtzia densa Benth.“
- Line 97: verify name! - Lephidalis holosteroides, edit „Lepyrodiclis holosteoides (C. A. Mey.) Fisch. et C. A. Mey.“
2.3. Test crops
- Lines 100 – 101: edit „Cucumis sativus L., Spinacia oleracea L., Lactuca sativa L. var. asparagina, Pisum sativum L., Hordeum vulgare L.,
Author Response
Dear Dr. Reviewer,
Thank you very much for your letter and we are very grateful for the constructive comments and valuable recommendations given by you. Based on your suggestions and edits, we have carefully revised the manuscript.
Here below is our Point-by-point responses to the comments.
Comments and Suggestions
1*. Abstract:
Line 21: edit „Cucumis sativus L., Lactuca sativa L. var. asparagina,“
Line 36: edit „Hordeum vulgare L.)“
Response1*. The Latin names that need to be modified in the abstract have been revised in accordance with the editor's suggestions.
2*. 2.2 Test weeds
Line 96: edit „Elsholtzia densa Benth.“
Line 97: verify name! - Lephidalis holosteroides, edit „Lepyrodiclis holosteoides (C. A. Mey.) Fisch. et C. A. Mey.“
Response2*. The Latin names that need to be modified in the Test weeds have been revised in accordance with the editor's suggestions.
3*. 2.3. Test crops
Lines 100 – 101: edit „Cucumis sativus L., Spinacia oleracea L., Lactuca sativa L. var. asparagina, Pisum sativum L., Hordeum vulgare L.,
Response3*. The Latin names that need to be modified in the Test crops have been revised in accordance with the editor's suggestions.
Great thanks to you and all the reviewers for the time and effort you expend on this manuscript. If the manuscript still needs to be revised, we will continue to work on it.
Reviewer 2 Report
Comments and Suggestions for Authors
No further comments
Reviewer 3 Report
Comments and Suggestions for Authors
please rename the section 4 as Discussion (otherwise there are two sections with Conclusion)